# Non-Classical HLA Determinants of the Clinical Response after Autologous Stem Cell Transplantation for Systemic Sclerosis

**DOI:** 10.3390/ijms23137223

**Published:** 2022-06-29

**Authors:** Wahid Boukouaci, Pauline Lansiaux, Nathalie C. Lambert, Christophe Picard, Emmanuel Clave, Audrey Cras, Zora Marjanovic, Dominique Farge, Ryad Tamouza

**Affiliations:** 1Translational Neuropsychiatry Laboratory, Institut National de la Santé et de la Recherche Médicale (IN-SERM, U955), Institut Mondor de Recherche Biomédicale, Université Paris Est Creteil, F-94010 Creteil, France; w_boukouaci@yahoo.fr; 2Unité de Médecine Interne (UF 04): CRMR MATHEC, Maladies Auto-Immunes et Thérapie Cellulaire, Centre de Référence des Maladies Auto-Immunes Systémiques Rares D’ILE-de-France, Hôpital St-Louis, Assistance-Publique Hôpitaux de Paris, F-75010 Paris, France; pauline.lansiaux@aphp.fr; 3URP-3518: Recherche Clinique en Hématologie, Immunologie et Transplantation, Institut de Recherche Saint-Louis, Université Paris Cité, F-75010 Paris, France; 4UMRs 1097 Arthrites Autoimmunes, Institut National de la Santé et de la Recherche Médicale (INSERM), Aix Marseille University, F-13288 Marseille, France; nathalie.lambert@inserm.fr; 5UMR7268 ADES (Anthropologie Bio-Culturelle, Droit, Ethique et Santé), Université Aix-Marseille, Etablissement Français du Sang, Centre National de la Recherche Scientifique (CNRS), F-13005 Marseille, France; christophe.picard@efs.sante.fr; 6EMiLy (Ecotaxie, Microenvironnement et Developpement Lymphocytaire), Inserm U1160, Institut de Recherche Saint Louis, Université de Paris, F-75010 Paris, France; emmanuel.clave@u-paris.fr; 7Cell Therapy Unit, Saint Louis Hospital, Assistance-Publique Hôpitaux de Paris, F-75010 Paris, France; audrey.cras@aphp.fr; 8UMR1140, Institut National de la Santé et de la Recherche Médicale (INSERM), Université de Paris, F-75006 Paris, France; 9Department of Hematology, Hopital Saint Antoine, F-75012 Paris, France; zora.marjanovic@aphp.fr; 10Department of Medicine, McGill University, Montreal, QC H3A 1A1, Canada; 11Fondation FondaMental, Département Médico-Universitaire de Psychiatrie et d’Addictologie (DMU IMPACT), Fédération Hospitalo-Universitaire de Médecine de Précision en Psychiatrie (FHU ADAPT), Assistance-Publique Hôpitaux de Paris, F-94010 Creteil, France

**Keywords:** systemic sclerosis (SSc), inflammation, autologous hematopoietic stem cell transplantation (AHSCT), responder status, HLA-G, HLA-E

## Abstract

Systemic Sclerosis (SSc) is a chronic autoimmune disease with high morbidity and mortality. Autologous Hematopoietic Stem Cell Transplantation (AHSCT) is the best therapeutic option for rapidly progressive SSc, allowing increased survival with regression of skin and lung fibrosis. The immune determinants of the clinical response after AHSCT have yet to be well characterized. In particular, the pivotal role of the Human Leukocyte Antigen (HLA) system is not well understood, including the role of non-classical immuno-modulatory HLA-E and HLA-G molecules in developing tolerance and the role of Natural Killer cells (NK) in the immunomodulation processes. We retrospectively tested whether the genetic and/or circulating expression of the non-classical HLA-E and HLA-G loci, as well as the imputed classical HLA determinants of HLA-E expression, influence the observed clinical response to AHSCT at 12- and 24-month follow-up. In a phenotypically well-defined sample of 46 SSc patients classified as clinical responders or non-responders, we performed HLA genotyping using next-generation sequencing and circulating levels of HLA-G and quantified HLA-E soluble isoforms by ELISA. The -21HLA-B leader peptide dimorphism and the differential expression level of *HLA-A* and *HLA-C* alleles were imputed. We observed a strong trend towards better clinical response in *HLA-E**01:03 or *HLA-G* 14bp Del allele carriers, which are known to be associated with high expression of the corresponding molecules. At 12-month post-AHSCT follow-up, higher circulating levels of soluble HLA-E were associated with higher values of modified Rodnan Skin Score (mRSS) (*p* = 0.0275), a proxy of disease severity. In the non-responder group, the majority of patients carried a double dose of the HLA-B Threonine leader peptide, suggesting a non-efficient inhibitory effect of the HLA-E molecules. We did not find any correlation between the soluble HLA-G levels and the observed clinical response after AHSCT. High imputed expression levels of HLA-C alleles, reflecting more efficient NK cell inhibition, correlated with low values of the mRSS 3 months after AHSCT (*p* = 0.0087). This first pilot analysis of HLA-E and HLA-G immuno-modulatory molecules suggests that efficient inhibition of NK cells contributes to clinical response after AHSCT for SSc. Further studies are warranted in larger patient cohorts to confirm our results.

## 1. Introduction

Systemic Sclerosis (SSc) is a rare and severe chronic systemic autoimmune disease characterized by early vascular alterations, activation of immune processes, and progressive fibrosis of skin and internal organs [1,2]. It is associated with reduced health-related quality of life (HRQoL) and a shorter life expectancy [3,4,5]. Early rapidly progressive SSc is the most lethal connective tissue disease, with a 5-year survival rate of 50–70% depending on the extent of organ involvement [6,7]. Risk factors for high mortality are diffuse skin fibrosis [7], clinically assessed by the modified Rodnan skin score (mRSS, range 0–51) [8], elevated C-reactive protein levels [9], altered left or right ventricular ejection fraction [10], interstitial lung disease with reduced forced vital capacity (FVC) or diffusion capacity for carbon monoxide (DLCO) on lung function tests [11], proteinuria, and male gender [9]. The presence of anti-topoisomerase II (Scl-70) auto-antibodies, reduced functional status on the Scleroderma Health Assessment Questionnaire (sHAQ) [12], and pulmonary arterial hypertension were also shown to adversely affect survival [7,13].

Three successive randomized controlled trial (RCT) demonstrated the efficacy of Autologous Hematopoietic Stem Cell Transplant (AHSCT) compared to cyclophosphamide intravenous (iv) pulses. ASSIST (American scleroderma stem cell versus immune suppression trial) [14] and ASTIS (Autologous stem cell transplantation international scleroderma trial) [15] used a non-myeloablative regimen and the SCOT trial (Scleroderma: Cyclophosphamide or transplantation) [16] used a myeloablative conditioning regimen. AHSCT has become the best therapeutic option for patients with severe or rapidly progressive SSc, and is recommended by the 2016 American Rheumatism Association (ACR)-European League Against Rheumatism (EULAR) guidelines (grade A level of evidence) [17], the American Society for Transplantation and Cellular Therapy (ASBMT) (standard of care) [18], and the European Bone Marrow Transplant Association (EBMT) (grade 1 level of evidence) [19].

Several immune processes may underpin the inter-individual responses to AHSCT in SSc patients. SSc patients can be categorized as responders (R) or non-responders/relapsing (NR), using pre-defined criteria systematically assessed during routine post-transplant follow-up [20,21]. Specific analysis of T and B cell reconstitution profiles and diversity of the T-cell repertoire, together with mRSS and FVC scoring at 1–2 years post-transplant has permitted the identification of long-term clinical responders and non-responders/relapsing SSc patients at 4–5 years after AHSCT [22].

Certain innate and adaptive immune processes are associated with the clinical evolution of SSc before and after AHSCT [23]. The non-classical HLA-class I molecules, specifically HLA-E and HLA-G, contribute to Natural Killer (NK) cell-mediated immunomodulatory function [24,25,26]. HLA-E and HLA-G exist either as membrane-bound molecules or as soluble circulating isoforms. Both HLA-E and HLA-G are encoded by pauci-polymorphic loci. HLA-E is encoded by two functional alleles, namely *HLA-E**01:01 and *HLA-E**01:03, which differ by a single amino acid substitution (Arginine to Glycine at position 107) in the alpha-2 heavy chain domain and by different cell surface expressions; *HLA-E**01:01 being almost undetectable, while *HLA-E**01:03 is expressed at normal levels [27,28]. The HLA-E soluble circulating isoform (sHLA-E) results from the shedding of membrane bound HLA-E molecules, which can be induced by stressful events, such as infection or inflammation. In addition, the rates of expression of both *HLA-A* and *HLA-C* classical alleles appear to modulate the HLA-E expression through allele-dependent release of the M leader peptide. Higher HLA-A and C expression levels are believed to yield higher expression levels of HLA-E molecules, given that all *HLA-A* and *C* alleles carry the methionine leader peptide [29,30]. HLA-G expression is characterized by seven isoforms consisting of four membrane-bound (HLA-G1 to -G4) and three soluble isoforms (sHLA-G) (HLA-G5 to -G7). *HLA-G* locus displays a unique polymorphism pattern distributed along the non-coding regions, namely the promoter and the 3′-untranslated (3′UTR) regions, with a limited number of exonic polymorphisms with 102 alleles and 35 protein variants reported so far (IMGT/HLA sequence database, March 2022). Each allele carries a 14-base pair (bp) insertion (Ins) or deletion (Del) in the 3′UTR which modulates HLA-G expression and a higher circulating level of sHLA-G is associated with the Del/Del genotype as compared to the Ins/Ins genotype [31].

We took advantage of a phenotypically well-defined sample of 46 SSc patients before and after AHSCT to retrospectively analyze the influence of the HLA-E and HLA-G circulating expression levels and genetic diversity on the observed clinical responses at different time points during follow-up after AHSCT for SSc.

## 2. Results

### 2.1. Socio-Demographical Characteristics and Response to AHSCT

The study population included 29 females and 17 males, with a mean age of 45.6 years (±12.8) [range: 17–66] at time of AHSCT. SSc patients were from European (58.7%), Afro-American (17.4%), North African (15.2%), and Asian (8.7%) geographic origins (Table 1). According to the observed clinical response after AHSCT, 32 patients were classified as responders (R) and 14 patients as non-responders/relapsing (NR) at 12 months. At 24 months, 33 patients were classified as responders (R) and 13 patients as non-responders/relapsing (NR) (Table 2). No statistically significant differences in clinical responses were observed according to sex, geographic origin, and age at time of transplant (Table 2 and Appendix A).

### 2.2. Non Classical and Classical HLA Class I Genetics and Response to AHSCT

We observed a trend towards a higher frequency of high *HLA-E**01:03 and *HLA-G* 14bp Del expressor alleles in responder patients. Overall, non-classical and classical HLA class I genetics analyses did not identify any statistically significant differences between responder and non-responder patients (Table 3, Appendix A).

HLA-E expression appeared lower in non-responder patients with higher frequency of the homozygous state of the low *HLA-C* expressor *rs*2395471 G allele and of the homozygous state of the T-HLA-B leader peptide allele in responders at 12 and 24 months after AHSCT. None of these results reached statistical significance (Table 3).

### 2.3. HLA-A and C and the Modified Rodnan Skin Score (mRSS)

A significant negative correlation between the imputed quantitatively low level of HLA-C expression and higher mRSS score was observed at baseline (*p* = 0.0075; r = −0.3937) and 3 months post-AHSCT (*p* = 0.0117; r = −0.3854), but this was no longer the case by 12 months post-AHSCT (*p* = 0.0654; r = −0.2802) (Figure 1). At baseline, the median mRSS in patients with the homozygous rs2395471AA *HLA-C* high expressor genotype was not significantly different compared to those with the rs2395471GG/AG genotype (19.5 vs. 25, *p* = 0.0674). By the 3-month follow-up, mRSS was significantly lower in patients homozygous for the rs2395471AA *HLA-C* allele (9 vs. 19, *p* = 0.0087) (Figure 2). This result remained significant when others variables (sex, geographic origin, age at transplant, disease duration, presence of interstitial lung disease (ILD) at baseline, and presence of anti-Scl70 antibodies at baseline) were taken into account in multivariate analyses (*p* = 0.041) (Appendix A).

We did not find any correlation between the imputed expression levels of HLA-A and mRSS scores (data not shown).

### 2.4. Circulating Levels of Soluble HLA-G and E Molecules

A significant positive correlation between sHLA-E levels and mRSS (*p* = 0.0275; r = 0.3624, Figure 3) was observed at 12 months post-AHSCT. The levels of sHLA-E at baseline were not predictive of mRSS evolution at 12 months (data not shown). We found that post-transplant clinical responses after AHSCT were not influenced by the circulating levels of sHLA-G molecules at baseline, nor during follow-up (data not shown).

## 3. Discussion

This retrospective analysis of 46 phenotypically well-defined SSc patient before and after AHSCT was focused on the genetic and/or the circulating expression of two potent immuno-modulatory molecules namely HLA-E and HLA-G, and their potential influence on clinical response.

In terms of function, upon binding to self-peptides from various HLA-class I molecules, the HLA-E molecules modulate the natural killer (NK) cell responses through interaction with the CD94-NKG2A inhibitory NK cell receptor, with consequent inhibition of NK cell-mediated cytotoxicity and cytokines production [27,32,33]. HLA-G molecules also play a prominent role in immune tolerance and are important immune checkpoints. They are tightly involved in the inhibition of NK cell cytotoxicity, antigen-specific cytotoxic CD8 + T cell functions, and CD4 + T cell allogeneic proliferation. Consequently, HLA-G molecules settings may influence post-AHSCT complications in autoimmune/rheumatologic diseases, including in SSc [34,35,36,37].

In the present study, we observed that imputed high expression levels of *HLA-C* alleles were associated with lower mRSS after AHSCT. Both the quantitative expression of *HLA-C* alleles per se and rs2395471 polymorphism were associated with lower mRSS after AHSCT. These results may reflect the relationship between NKG2A-HLA-E and killer cell immunoglobulin-like (KIR)-HLA and NK cell inhibition. High expression of *HLA-C* alleles increases available M leader peptide thereby allowing more efficient expression of HLA-E and consequent NK cell inhibition. High expression of HLA-C also contributes to inhibiting NK cell activation via a different mechanism involving interaction with the inhibitory KIR receptors [38]. The ontogeny of NK cells is characterized by sequential early to late developmental stages with the respective expression of specific CD56^bright^ or CD56^dim^ cell surface molecules. While the early CD56^bright^ NK cell uses the CD94/NKG2A receptor to interact with HLA-E molecules towards mediating NK cell inhibition, NK cell maturation is accompanied by a shift from NKG2A to KIR-mediated NK cell inhibition [38,39]. Hence, one can hypothesize that: (i) clinical response to AHSCT may be related, at least in part, to NK cell inhibition and (ii) such processes are likely to involve sequential HLA-E/NKG2A- and HLA-C/KIR-mediated interactions following NK cell maturation during post-transplant immune reconstitution process [23].

Given the above-mentioned HLA-E characteristics, analysis of the potential influence of the HLA-B T leader peptide dimorphism failed to provide significant associations even though the HLA-B T leader peptide was predominantly found in non-responder patients suggesting an inefficient HLA-E-mediated NK cell inhibition with sustained inflammatory processes in this setting after AHSCT for SSc.

We also observed a trend towards a better response in patients bearing either the high expression *HLA-E**01:03 allele or the *HLA-G* high expressor 14 bp Del allele, suggesting that the ability of both HLA-E and HLA-G molecules to inhibit effector cells (NK and or T cells), after AHSCT, is genetically determined. Future studies with larger numbers of SSc patients undergoing AHSCT are warranted to confirm the present exploratory results, to be in line with previous findings after allogeneic HSCT [40,41,42,43].

High circulating levels of sHLA-E were associated with high mRSS, and constitute a potential risk factor for non-response after AHSCT in SSc patients. This has been observed in patients with acute and chronic Graft versus Host Disease after allogenic HSCT [44]. High circulating levels of sHLA-E were also found in various cancer settings [44,45,46,47], which led to the development of a novel immune check point inhibitor targeting the HLA-E specific NKG2A inhibitor receptor, namely Monalizumab [47].

We did not find any correlation between sHLA-G levels and post AHSCT outcomes, an observation that is somewhat intriguing in the light of numerous studies in organ or stem cell transplantation, showing that high sHLA-G levels are often associated with better post-transplant outcomes [35,48]. In SSc patients, higher expression of both HLA-G membrane bound and soluble isoforms were observed compared to healthy controls [49,50]. Other studies showed that low sHLA-G levels were associated with severe forms of SSc diseases. These controversial data together with our present findings may also reflect a more complex regulation of SSc-related sHLA-G synthesis after treatment by AHSCT. However, we cannot exclude a simple lack of statistical power due to the relatively low sample size.

This descriptive study presents several limitations and may be underpowered due to the small sample size (46 SSc patients). However, this small, but unique sample study population is of importance. Systemic Sclerosis is a rare chronic auto-immune disease (prevalence 15/100,000 people), with the highest morbidity and mortality of all rheumatic diseases. To date, AHSCT is the only treatment to improve overall survival and event-free survival in severe rapidly progressive diffuse SSc patients [15,20]. During the study period, 66 SSc patients were transplanted in France overall. Therefore, analyzing the present cohort of 46 SSc patients treated by AHSCT, with appropriate sample collection, has generated important preliminary data on the role of HLA-G and HLA-E in the immune response following AHSCT in SSc patients. Another study limitation is the lack of functional experiments in this retrospective study. Only a few vials of frozen peripheral blood mononuclear cells (PBMCs) preserved in Dimethyl sulfoxide (DMSO) were available for the study and were used for DNA extraction dedicated to genetic analysis, the primary endpoint of the study.

In summary, this study first addressed the potential influence of genetic and expression profile of two potent immunomodulatory molecules, namely HLA-E and HLA-G, on clinical responses following AHSCT for SSc. If confirmed/replicated, our observations may lead to a better understanding and management of clinical responses to transplantation not only for SSc but for other systemic autoimmune disorders.

## 4. Material and Methods

### 4.1. Study Subjects

Forty-six SSc patients, diagnosed according to the 2013 ACR/EULAR criteria [51], were treated with autologous stem cell transplant (AHSCT) between 2002 and 2018 at either the Saint-Louis or at Saint Antoine hospitals (Assistance-Publique Hôpitaux de Paris (AP-HP)), using the ISAMAIR (Intensification et Autogreffe dans les Maladies Auto Immunes Résistantes, phase I-II) [20] or the ASTIS (Autologous stem cell transplantation international scleroderma trial, phase III) [15] clinical trials protocols, or routine AHSCT care procedures thereafter (www.mathec.com, accessed on 28 June 2022). All patients provided informed consent and were included in the prospective Maladies Auto-Immunes et Therapie Cellulaire (MATHEC) cohort for long-term follow-up, data collection, and analysis.

Clinical follow-up was performed according to the European [52] and the French [53] good clinical practice guidelines for patient evaluation and monitoring. Patients were assessed before AHSCT and followed for two years post-transplant, on a quarterly basis during the first year post-transplant and twice a year thereafter. According to the observed clinical response at 12 or 24 months after AHSCT, SSc patients were retrospectively classified as either responders (R) or non-responders (NR), as previously published [20,21]. Response to treatment was defined as (1) at least 25% improvement of mRSS or (2) greater than 10% increase in Forced Vital Capacity (FVC) and/or in pulmonary diffusion capacity for carbon monoxide (DLCO) as compared to the baseline scores. Non-responders were patients with SSc progression or relapse at the time of evaluation. Progression and Relapse were defined by any of the following criteria when comparing the best observed response to baseline: at least a 25% increase in mRSS or a 10% decrease in FVC and/or 15%, decrease in DLCO, onset of renal crisis, start of total parenteral nutrition, or need for new immune-suppressive or modulating medication after AHSCT [20,21].

Pre- and post-transplant peripheral blood mononuclear cells (PBMCs), plasma, and serum, were collected at baseline before the mobilization and at 3, 6, and 12 months after AHSCT, as well as semi-annually during the second year. The samples were immediately processed and stored at required temperatures until experiments according to EBMT [54] and MATHEC-SFGM-TC [55] guidelines. The study was conducted in accordance with the Declaration of Helsinki and Good Clinical Practice guidelines. All patients provided written informed consent for research on their biological material and clinical data.

### 4.2. HLA Genotyping

Genomic DNA was extracted from EDTA-treated peripheral blood samples or from frozen peripheral blood mononuclear cells (PBMCs) using commercially available kits (Qiagen EZ1 blood or Tissue kit, respectively, Qiagen, Hilden, Germany) and were quantified by spectrophotometry analysis (BioDrop µLITE+, Biochrom, Cambridge, UK).

Classical *HLA* class I (*A, B, C*) and non-classical *HLA* class I (*E, F, G*) loci genotyping was performed on the DNA samples from the 46 SSc patients at the Immunogenetic laboratory of the French Blood Transfusion Department (EFS), Marseille, France using Next-Generation Sequencing (NGS, NG mix) technology. The 3’UTR-*HLA-G* haplotypes were reconstructed, using an EM algorithm from the Gene [56] program and confirmed using the EM and ELB algorithms from the Arlequin v3.5.1.2 package. Data were analyzed and interpreted as previously described [57,58].

### 4.3. Soluble HLA-G and E Measurements

The levels of circulating sHLA-G and sHLA-E were measured on plasma and serum samples collected before AHSCT (baseline), and at 3 and 12 months (±2 months) post-transplant during routine follow-up.

Both soluble HLA-G1 and soluble HLA-G5 isoforms were measured using the enzyme-linked immunosorbent assay (ELISA) kit (EXBIO/Biovendor, Karásek, Czech Republic; capture antibody: MEM-G/9), defined at the “Wet Workshop for the Quantification of sHLA-G” in 2004 [59], according to the manufacturer’s instructions. Soluble HLA-G standard was diluted to obtain a calibrator curve within a range from 3.91 to 125 international units/mL (IU/mL) for sHLA-G ELISA. The total protein concentration levels were expressed in IU/mL of plasma.

Circulating sHLA-E levels were measured using a dedicated ELISA with sandwich enzyme immunoassay for in vitro quantitative measurement of sHLA-E according to the manufacturer’s recommendations (cloud-clone corp. Katy, TX, USA).

### 4.4. Imputation of Classical HLA Determinants of HLA-E Expression

To examine whether functionally important classical HLA variations, known to influence HLA-E expression and consequently NK cell functions, have an impact on the treatment response status, we performed high resolution genotype imputation to determine: (i) the -21 exon 1 HLA-B methionine (M) to threonine (T) leader peptide change that categorize patients as strong or weak expressor of HLA-E molecules respectively and (ii) the differential expression level of both *HLA-A* and *HLA-C* alleles by determining the quantitative expression of *HLA-A* and *HLA-C* alleles per se [29,30] or the rs2395471 polymorphism, given that higher HLA-A and HLA-C expression levels result in higher expression of HLA-E molecules.

### 4.5. Statistical Analyses

Chi-square “χ^2^” tests (or Fisher test for small groups) were used to test whether gender, age, and geographic origin, HLA haplotype, or HLA polymorphism frequencies varied between responders (R) and non-responders (NR).

A Mann–Whitney test was applied to compare sHLA-G or sHLA-E levels/concentrations between responder (R) and non-responder (NR) patients, independently or according to *HLA-G/E* genetic status, and to compare the evolution of mRSS between patients stratified according to their classical and non-classical HLA genotype.

Spearman correlation was applied to assess the relationship between mRSS at baseline and at 3, 6, and 12 months post-AHSCT, and soluble HLA-G and HLA-E levels or HLA-A and HLA-C imputed expression levels. *p*-values less than 0.05 were considered significant (GraphPad Prism 6 (La Jolla, CA, USA).

A linear regression model was used to evaluate the relationship between sex, geographic origin, age at transplant, disease duration, presence of interstitial lung disease (ILD) on High-resolution computed tomography (HRCT) at baseline, presence of anti-Scl70 antibodies at baseline, and *HLA-C* rs2395471, as independent variables, with the mRSS value at 3 months after AHSCT as a continuous dependent variable, using R software version 4.0.3. *p*-values less than 0.05 were considered significant.

## Figures and Tables

**Figure 1 ijms-23-07223-f001:**
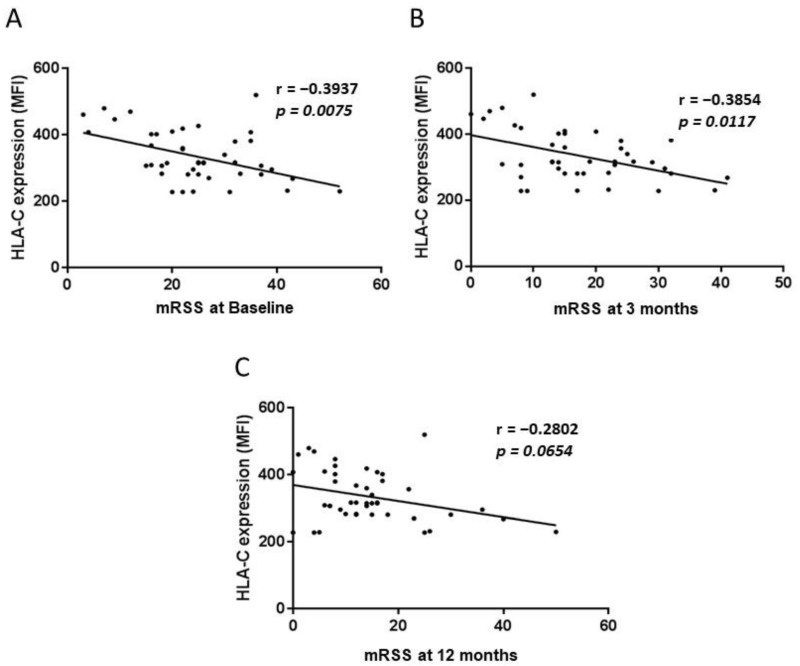
Correlation between the imputed quantitative HLA-C expression and modified Rodnan Skin Scores (mRSS) in Systemic Sclerosis patients before autologous hematopoietic stem cell transplantation (baseline) (**A**), and at 3- (**B**) and 12-month (**C**) follow-up.

**Figure 2 ijms-23-07223-f002:**
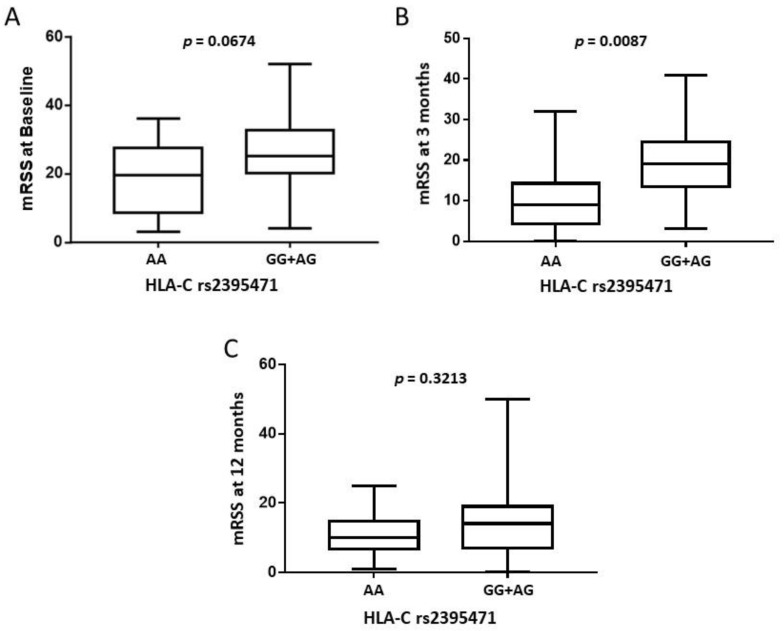
Comparison between the imputed *HLA-C* rs2395471 genotype and modified Rodnan Skin Scores (mRSS) in Systemic Sclerosis patients before autologous hematopoietic stem cell transplantation (baseline) (**A**), and at 3- (**B**) and 12-month (**C**) follow-up.

**Figure 3 ijms-23-07223-f003:**
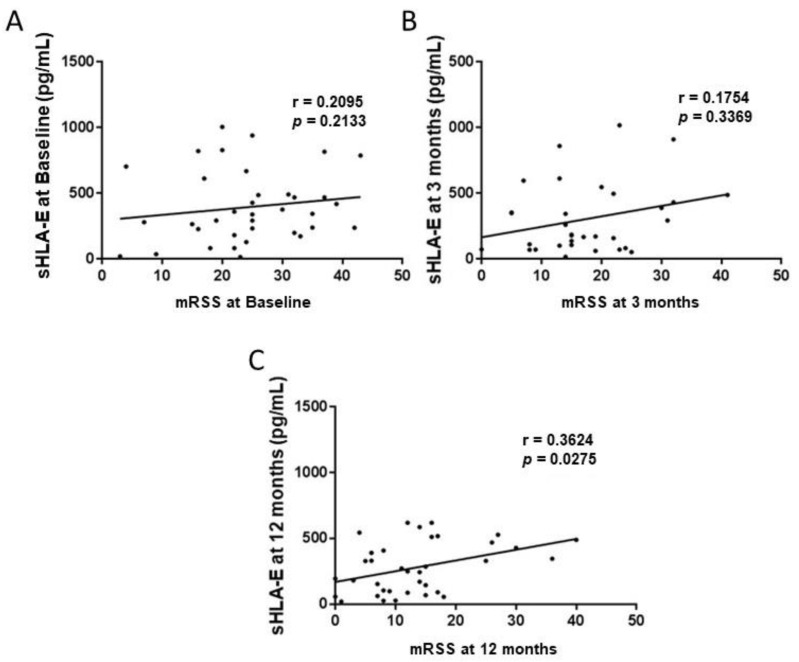
Correlation between the circulating levels of soluble HLA-G and HLA–E and Rodnan Skin Scores (mRSS) in Systemic Sclerosis patients before autologous hematopoietic stem cell transplan-tation (baseline) (**A**), and at 3- (**B**) and 12-month (**C**) follow-up.

**Table 1 ijms-23-07223-t001:** Demographic and clinical characteristics of 46 Systemic Sclerosis patients before autologous hematopoietic stem cell transplantation (AHSCT).

	*n* (%)/Mean (SD) [Min–Max]	Missing Values (*n*)
Age, years	45.6 (12.8) [17–66]	
Sex		
Male	17 (37.0)	
Female	29 (63.0)	
Geographic origin		
European	27 (58.7)	
Afro-American	8 (17.4)	
North African	7 (15.2)	
Asian	4 (8.7)	
Disease duration, years	2.3 (1.5) [0.2–6.2]	
modified Rodnan Skin Score (mRSS)	24.9 (10.3) [3–51]	
Pulmonary involvement		
FVC, % predicted	78.0 (17.2) [52–130]	2
DLCO, % predicted	52.6 (15.7) [26–90]	
Auto-antibodies		
Antinuclear positive	44 (95.7)	1
Anti-Scl 70 positive	29 (63.0)	1
Anti-centromere positive	1 (2.2)	2

DLCO: Diffusing capacity of the Lungs for Carbon Monoxide; FVC: Forced Vital Capacity; SD: standard deviation. Continuous variables are summarized as mean (Standard deviation) [minimum-maximum] and categorical variables as numbers of patients (percentage).

**Table 2 ijms-23-07223-t002:** Demographic characteristics of 46 Systemic Sclerosis patients, according to the observed clinical response (responder or non-responder) at 12 and 24 months after autologous hematopoietic stem cell transplantation (AHSCT).

	12 Months after AHSCT				24 Months after AHSCT			
	R, *n* (%)	NR, *n* (%)	*p*-Value R vs. NR	Global *p*-Value	R, *n* (%)	NR, *n* (%)	*p*-Value R vs. N	Global *p*-Value
	*n* = 32	*n* = 14			*n* = 33	*n* = 13		
Geographic origin								
European	20 (62.5)	7 (50)	0.43		20 (60.6)	7 (53.8)	0.68	
Afro-American	4 (12.5)	4 (28.6)	0.19		4 (12.1)	4 (30.8)	0.20	
North African	6 (18.8)	1 (7.1)	0.41		6 (18.2)	1 (7.7)	0.65	
Asian	2 (6.2)	2 (14.3)	0.37		3 (9.1)	1 (7.7)	1.00	
European and North African	26 (81.3)	8 (57.1)		0.16	26 (78.8)	8 (61.5)		0.23
Afro-American and Asian	6 (18.8)	6 (42.9)			7 (21.2)	5 (38.5)		
Age at transplantation								
Mean (SD),	45.6 (12.4)	45.6 (14.4)	0.44		45.4 (13.0)	45.9 (12.9)	0.86	
Median	45.8	50.6			46.8	48.9		
Min–Max	20.1–66.4	16.7–61.5			20.1–66.4	16.7–61.1		
Sex								
Male	14 (43.8)	3 (21.4)		0.15	14 (42.4)	3 (23.1)		0.32
Female	18 (56.3)	11 (78.5)			19 (57.6)	10 (76.9)		

AHSCT: Autologous Hematopoietic Stem Cell Transplantation; R: responder patients, NR: non-responder patients, SD: Standard Deviation.

**Table 3 ijms-23-07223-t003:** Genotype distributions and allele frequencies of *HLA-E* and *G* and the imputed HLA-B and C polymorphisms in Systemic Sclerosis patients, according to the observed clinical response (responder or non-responder) 12 and 24 months after autologous hematopoietic stem cell transplantation (AHSCT).

	12 Months after AHSCT				24 Months after AHSCT			
	R, *n* (%)	NR, *n* (%)	*p*-Value R vs. NR	Global *p*-Value	R, *n* (%)	NR, *n* (%)	*p*-Value R vs. NR	Global *p*-Value
*HLA-E* genotype	*n* = 32	*n* = 12			*n* = 32	*n* = 12		
E*01:01/E*01:01	7 (21.9)	3 (25)	1.00		7 (21.9)	3 (25)	1.00	
E*01:01/E*01:03	18 (56.2)	9 (75)	0.26		18 (56.2)	9 (75)	0.26	
E*01:03/E*01:03	7 (21.9)	0 (0)	0.16		7 (21.9)	0 (0)	0.16	
*HLA-E* allele	*n* = 64	*n* = 24			*n* = 64	*n* = 24		
E*01:01	32 (50)	15 (62.5)		0.30	32 (50)	15 (62.5)		0.30
E*01:03	32 (50)	9 (37.5)			32 (50)	9 (37.5)		
*HLA-G* 14 bp genotype	*n* = 32	*n* = 13			*n* = 33	*n* = 12		
DEL/DEL	13 (40.1)	3 (23.1)	0.32		14 (42.4)	2 (16.7)	0.16	
INS/DEL	13 (40.1)	7 (53.8)	0.42		13 (39.4)	7 (58.3)	0.26	
INS/INS	6 (18.8)	3 (23.1)	0.70		6 (18.2)	3 (25)	0.68	
*HLA-G* 14 bp allele	*n* = 64	*n* = 26			*n* = 66	*n* = 24		
DEL	39 (60.9)	13 (50)		0.34	41 (62.1)	11 (45.8)		0.17
INS	25 (39.1)	13 (50)			25 (37.9)	13 (54.2)		
*HLA-C* rs2395471 genotype	*n* = 32	*n* = 13			*n* = 33	*n* = 12		
AA	6 (18.8)	4 (30.8)	0.44		6 (18.2)	4 (33.3)	0.42	
AG	21 (65.6)	6 (46.2)	0.23		23 (69.7)	4 (33.3)	0.04	
GG	5 (15.6)	3 (23.1)	0.67		4 (12.1)	4 (33.3)	0.18	
*HLA-C* rs2395471 allele	*n* = 64	*n* = 26			*n* = 66	*n* = 24		
A	33 (51.6)	14 (53.8)		0.84	35 (53)	12 (50)		0.80
G	31 (48.4)	12 (46.2)			31 (47)	12 (50)		
HLA-B peptide leader genotype	*n* = 32	*n* = 13			*n* = 33	*n* = 12		
MM	3 (9.4)	0 (0)		0.19	3 (9.1)	0 (0)		0.53
TM	8 (25.0)	8 (61.5)			10 (30.3)	6 (50)		
TT	21 (65.6)	5 (38.5)			20 (60.6)	6 (50)		

AHSCT: Autologous Hematopoietic Stem Cell Transplantation; R: responder patients, NR: non-responder patients, SD: Standard Deviation. Due to missing DNA samples, one non-responder patient was not typed for all HLA loci and DNA from one non-responder was insufficient for HLA-E genotyping. A total of 13 patients and 12 patients were analyzed for the respective HLA-G and HLA-E genotyping calculations among the 14 NR patients at 12 months. In total, 12 patients were analyzed for HLA-G and HLA-E genotyping calculations among the 13 NR patients at 24 months.

## Data Availability

The data presented in this study are available on reasonable request from the corresponding author.

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
