# Peer review of "Non-Classical HLA Determinants of the Clinical Response after Autologous Stem Cell Transplantation for Systemic Sclerosis"

_ijms, 2022, doi:10.3390/ijms23137223_

Round 1
Reviewer 1 Report
- This descriptive study presents several limitations (including the small sample size investigated and the lack of functional experiments) that should be adequately acknowledged and discussed by the authors. Indeed, the Discussion section lacks a "study limitations" paragraph.
- Due to the preliminary, but significant, nature of the findings, the study should be better presented as "Communication" (not as Full Article).
- The Results section contains lengthy description of many non-significant findings that should be better summarized in Tables.
- Findings with p = 0.06 cannot be referred to as statistically significant. Please, carefully check the Results section.
- A few functional experiments using PBMCs from responders and non-responders SSc patients should be implemented in order to support the authors' speculations on NK cells.
- Please, carefully check the entire text for typos and some grammatical errors (for instance, "data not showed" should be "data not shown").
Reviewer 2 Report
 In this article, authors investigate to identify predictors of Autologous Hematopoietic Stem Cell Transplantation (AHSCT) treatment responders. The immune determinants of the clinical response after AHSCT are yet unraveled, specifically the pivotal role of the Human Leukocyte Antigen (HLA) system, which includes the non-classical immuno-modulatory HLA-E and HLA–G molecules involved in tolerance and Natural Killer cells (NK) immunomodulation. Authors retrospectively tested whether the non-classical HLA-E and HLA-G loci genetic and/or circulating expression status, as well as the imputed classical HLA determinants of HLA-E expression, influence the observed clinical response after AHSCT at 12 or 24 months. Among a phenotypically well-defined sample of forty-six SSc patients, classified as clinical responders or not, HLA genotyping was performed using next generation sequencing and circulating levels of HLA-G and HLA-E soluble isoforms were quantified by ELISA methods. The HLA-B leader peptide dimorphism and the differential expression level of HLA-A and HLA-C alleles were imputed. We observed a strong trend towards a better clinical response in HLA-E*01:03 or HLA-G 14bp Del allele carriers, which are known to be associated with high expression of the corresponding molecules. At 12 months after AHSCT, higher circulating levels of soluble HLA-E were associated with higher values of modified Rodnan Skin Score (mRSS) (p=0.0275), a proxy of disease severity. In the non-responder group, the majority of patients carried a double dose of the HLA-B Threonine leader peptide, suggesting a non-efficient inhibitory effect of the HLA-E molecules. They did not find any correlation between soluble HLA-G levels and the clinical response after AHSCT. High imputed expression levels of HLA-C alleles correlated with low values of the mRSS at 3 months after AHSCT (p=0.0087), reflecting more efficient NK cell inhibition. This first pilot analysis of the influence of the HLA E and G potent immuno-modulatory molecules suggests that efficient inhibition of NK cells may improve the clinical response after AHSCT for SSc. While the content is very interesting, I have some questions.
1) In table 2, you found no difference between Responder and Non-responder of AHSCT by age, gender, or race. But did background factors such as antibody type, presence of ILD, etc. make a difference?
2) In table 2 and figure 2, similarly, the mRSS of HLA-C rs2395471 allyl improved earlier for AA and GG+AG, whereas it improved more slowly for GG+AG. Do these differ by background factors such as gender, antibody type, ILD status, or race? That is, mRSS improved earlier for one antibody type, whereas mRSS improved more slowly for another antibody type, and so on. Also, is there any difference depending on the duration of the disease (e.g., AHSCT is less effective in those with a longer duration of disease)? If biopsies are taken, is it possible that the findings of the skin biopsy may indicate slow improvement if fibrosis is predominant, or early improvement if inflammation is predominant?
3) In this study, the progress is described only for mRSS, but what about the effect of AHSCT on digital ulcers? If effective, what were the results in this cohort? Does the type of HLA make it more or less effective? Is there a difference in the degree of improvement depending on the site where the ulcer is seen?
4) Similarly, what about the effect of AHSCT on ILDs? If effective, does the HLA type make a difference, e.g., early or slow improvement? Are there any other differences in the therapeutic effect of AHSCT based on the type of antibody or the pattern on HRCT of ILD (e.g., NSIP or UIP pattern)?
Round 2
Reviewer 1 Report
The authors successfully revised the manuscript taking into account the majority of my suggestions. Some additional experiments could not be performed, but their explanations are fully understandable.